# Barriers to and facilitators of the identification, management and referral of childhood anxiety disorders in primary care: a survey of general practitioners in England

Doireann O'Brien, Kate Harvey, Cathy Creswell

School of Psychology & Clinical Language Sciences, University of Reading, Reading, UK

**Correspondence to**
Dr Kate Harvey;
k.n.harvey@reading.ac.uk

## ABSTRACT

**Objectives** Although anxiety disorders are the most common emotional disorders in childhood and are associated with a broad range of negative outcomes, only a minority of affected children receive professional support. In the UK, general practitioners (GPs) are seen as 'gate-keepers' to mental health services. The aim of this study was to examine the extent to which GPs experience barriers to and facilitators of identifying, managing and accessing specialist services for these disorders, as well as factors associated with GPs' confidence.

**Design and setting** Cross-sectional, self-report questionnaire in primary care, addressing identification, management and access to specialist services for children (under 12 years) with anxiety disorders.

**Participants** 971 GPs in England.

**Primary outcomes** The primary outcomes for this research was the extent to which GPs felt confident (1) identifying and (2) managing anxiety disorders in children.

**Results** Only 51% and 13% of GPs felt confident identifying and managing child anxiety disorders, respectively. A minority believed that their training in identification (21%) and management (10%) was adequate. Time restrictions inhibited identification and management, and long waiting times was a barrier to accessing specialist services. Being female (Ex(B)=1.4, 95% CI 1.1 to 1.9) and being in a less deprived practice (Ex(B)=1.1, 95% CI 1 to 1.1) was associated with higher confidence identifying childhood anxiety disorders. Being a parent of a child over the age of 5 (Ex(B)=2, 95% CI 1.1 to 3.5) and being in a less deprived practice (Ex(B)=1.1, 95% CI 1 to 1.2) was associated with higher confidence in management. Receipt of psychiatric or paediatric training was not significantly associated with GP confidence.

**Conclusions** GPs believe they have a role in identifying and managing childhood anxiety disorders; however, their confidence appears to be related to their personal experience and the context in which they work, rather than their training, highlighting the need to strengthen GP training and facilitate access to resources and services to enable them to support children with these common but debilitating conditions.

---

**Strengths and limitations of this study**

► A key strength of the study is the inclusion of a large, nationally representative group of general practitioners (GPs).
► Findings from a recent systematic review, qualitative study and GP feedback informed the development of the questionnaire, ensuring that the most pertinent questions were asked and that the questions were meaningful to the GPs.
► The questionnaire distinguished between different stages of the primary care process which allowed pinpointing of the stage at which issues are most problematic.
► No data were available for GPs who chose not to participate and the possibility of self-selection biases influencing the findings cannot be excluded.
► The possibility that, in some cases, GPs answered questions in relation to mental health disorders other than anxiety cannot be ruled out.

## INTRODUCTION

Anxiety disorders are the most common mental health condition,[1] with a lifetime prevalence of 28.8%[2] and a median age of onset of 11 years.[2] As such, they are the most common emotional disorders in childhood,[3 4] with worldwide prevalence rates of 6.5% in children and adolescents[5] and are associated with an increased risk of subsequent mental health problems, substance abuse and poor educational attainment.[6 7] This level of impairment results in a high economic and societal burden,[8] with anxiety disorders being the leading cause of years lived with disability.[9] As such, there is a clear need for early identification and access to evidence-based interventions.[10] Effective treatments for childhood anxiety disorders exist[11 12]; however, only a minority of affected children access this support, with lower rates of treatment access than externalising disorders (such

as attention deficit hyperactivity disorder or conduct disorder) and mood disorders.[13 14]

In the UK, two-thirds of children see their general practitioner (GP) at least once a year[15] and there has been a steady rise in the number of children and young people presenting in primary care with mental health difficulties.[16 17] GPs are often the first medical professional that families see when they have mental health concerns and are in a position to develop strong relationships with families, in a non-stigmatising setting[18]. As such, they are often seen as 'gate-keepers' to accessing mental health services.[19 20] However, many GPs report a lack of confidence in their competence and skills in child and adolescent mental health, feel that they need further training[21] and believe that their role in this area requires further research and definition.[22] In addition, they face barriers such as insufficient resources when managing mental health conditions in primary care and extensive waiting lists for specialist services[23] and a high rate of rejected referrals. The reason for this is unclear, possibly reflecting issues with GP detection of mental health problems in this age group and lack of clarity regarding thresholds for Child and Adolescent Mental Health Services (CAMHS) among GPs, potentially associated with a lack of decision making aids for child and adolescent mental health disorders.[24]

The identification and management of childhood anxiety disorders may present a particular challenge for GPs due to their broad clinical presentation, frequent comorbidity with other mental health problems[2] and the common reliance on parents/caregivers to seek help.[25] Indeed, anxiety disorders are known to be less likely to be recognised by primary healthcare professionals than other childhood mental health problems,[26] potentially due to anxiety disorders being seen as less burdensome for the parents than externalising disorders.[27] In a recent qualitative study, there was wide variation in GPs' confidence managing anxiety disorders in (preadolescent) children, with many GPs describing feeling ill-equipped to manage and support children with these disorders.[28] While many of the barriers they described were similar to those experienced by GPs in relation to other mental health problems in children (eg, time constraints, difficulties accessing specialist help), GPs highlighted particular difficulties distinguishing anxiety disorders from physical conditions. It was also notable that they attributed their confidence in managing childhood anxiety disorders not only to factors such as training background and practice characteristics but also to whether they were a parent themselves.

Given the substantial burden that anxiety disorders pose in childhood, and beyond, successful early identification and appropriate management is imperative. However, as previous research has shown, GPs face numerous internal and external barriers in dealing with mental health disorders, with some early indication that anxiety disorders may bring specific challenges. This research aims to build on the findings of the previous qualitative work[28] to

ascertain a more representative and generalisable view of these issues throughout England. In addition to this, the study aims to develop an understanding of factors that influence GP confidence in dealing with these disorders.

## Research questions

RQ1: (1) What are the barriers and facilitators to (a) identification, (b) management and (c) referral to specialist services for preadolescent children with anxiety disorders for GPs in England?

RQ2: (2) What practice and personal factors are associated with GP confidence in identifying and managing childhood anxiety disorders?

## METHODS
### Procedure

Ethical approval for this study was granted.

Participants completed a 50-item self-report online questionnaire (approximately 15 minutes long) which contained three sections, addressing (a) identification, (b) management and (c) access to specialist services for children under 12 years with a suspected anxiety disorder (see online appendix 1). Free-text boxes for GPs to provide any additional barriers or comments were included at the end of each section and at the end of the survey. The content of the survey was informed by findings from a preceding systematic review[23] and qualitative study[28] which highlighted the utility of separating the stages of the primary care process. The questions were developed through an iterative process incorporating feedback from an academic GP and the research team.

The first half of each section contained questions about the GP's confidence and beliefs (eg, 'I feel confident identifying anxiety disorders in children under 12 years'). Based on the previous qualitative research, endorsement of these items indicated facilitators (and is labelled as such in the results section). The second half of each section asked GPs to rate external barriers they faced (eg, 'language barriers'). Responses to all items were rated on a five-point Likert scale (eg, from 'completely disagree' to 'completely agree', or 'not at all' to 'very much').

GPs provided information on gender, age, number of years qualified, whether they had completed psychiatric or paediatric training, whether they were a parent (including child's age) and whether they considered themselves 'research active'. Information on their practice population, such as size, socioeconomic status and ethnic diversity was obtained via their practice postcode.

## PARTICIPANTS

A priori sample size estimates (https://www.qualtrics.com/blog/calculating-sample-size/) indicated that at least 381 GPs were required to estimate the frequency of particular barriers with 95% CI, based on a total population of approximately 41 985 GPs in England,[29] and to

allow for robust analyses using a logistic regression with 10 predictor variables.[30]

Participants were GPs currently working in the National Health Service. There were no exclusion criteria. GPs were recruited through the National Institute for Health Research Clinical Research Networks (CRNs). Recruitment methods varied by CRN, from mass emails to all registered GPs in the region, to more targeted recruitment, including talks at GP forums. Multiple GPs per practice were allowed. Recruitment took place from 15 July 2016 to 6 February 2017 and covered 11 CRNS (out of a possible 15). CRNs contacted GPs within their regions via an email which contained a link to the survey. Paper copies were available on request.

Participation in this study was anonymous, although GPs' surgery postcode was requested to retrieve demographic information about the practice, and for CRN recruitment purposes. All participants were required to give informed consent.

## Patient and public involvement
No patients were involved in this research. One academic GP was involved in piloting the questionnaire.

## Statistical methods
Statistical analyses were performed using IBM SPSS (V.22).[31] As there was little missing data, cases which had missing data were excluded from analysis (number of cases missing per variable ranged between 7 and 20). To address research question one, frequencies, including CI, are reported for items on the questionnaire. To address research question two, logistic regression analyses were conducted with the dependent variables of confidence (1) identifying and (2) managing anxiety disorders in children under 12 years. These dependent variables were derived from the questions: 'I am confident identifying anxiety disorders' and 'I am confident managing anxiety disorders'. These questions were transformed into a binary format by collapsing 'agree' and 'completely agree' together and 'neutral', 'disagree' and 'completely disagree' together. The following predictors were included: patient population size (available on Public Health England), deprivation decile of practice (0–10, with 10 being the least deprived) (available on Public Health England), ethnicity of patient population (% non-Black Asian Minority Ethnic) (available on Public Health England), psychiatric/paediatric rotation completed as part of GP training (self-report), gender (self-report), number of years since qualification as a GP (self-report) and whether they were a parent of a child over the age of 5 years (self-report) (ie have personal experience of parenting a school-aged child). These factors were chosen based on previous qualitative research which indicated the potential influence they have on a GP's confidence.

The free-text comments were content-coded and organised into broad categories.

| Table 1 | Characteristics of general practitioners (GPs) and their practices | |
|---|---|---|
| **Variable** | | **Missing** |
| GP | | |
| % female | 52.70% | 9 |
| Number of years qualified | M=14.9 (range: <1–43) | 7 |
| Psychiatric rotation | 47.30% | 16 |
| Paediatric rotation | 73.50% | 11 |
| Child over 5 years old | 72% | 9 |
| Referred a child to specialist services for an anxiety disorder more than five times in the last 5 years | 58.8 | |
| Practice | | |
| Number of registered patients | M=12 009 (range: 1503–55499) | 19 |
| Socio-economic status of patients (deprivation decile of practice) | M=6.7 (range: 1–10) | 19 |
| % non-white ethnic groups | M=7% (range:. 8–72.2) | 20 |

## RESULTS
A total of 971 GPs completed the questionnaire. See table 1 for demographic information for a breakdown of participant numbers by CRNs.

i. What are the barriers and facilitators to the identification, management and referral of children, under the age of 12 years, with anxiety disorders for GP in the UK?

Table 2 reports the facilitators and barriers to identification, management and referral, as reported by GPs. A composite of those who agreed and strongly agreed, as well as those who disagreed and strongly disagreed, is presented for each item, in addition to the 95% CI.

ii. What factors are associated with a GP's confidence identifying managing this disorder?

A. The regression model (see table 3) was a significant fit overall ($\chi^2$ (1)=16.7, p<0.05). Being female was significantly associated with higher confidence identifying anxiety disorders in children, (Ex(B)=1.4, 95% CI 1.1 to 1.9), as was being in a less deprived practice (Ex(B)=1.1), 95% CI 1 to 1.1. GP psychiatric and paediatric training, as well as years of experience and parenthood did not have any impact on their confidence, nor did size or ethnicity of their practice population.

ii. What factors are associated with a GP's confidence managing this disorder?

B. The regression model (see table 4) was significant overall ($\chi^2$(1)=18.2, p<0.05.). Having a child over the age of 5 was significantly associated with greater confidence managing anxiety disorders in children (Ex(B)=2), 95% CI 1.1 to 3.5, as was being in a less deprived practice (Ex(B)=1.1, 95% CI 1 to 1.2). GP psy-

**Table 2** General practitioner (GP) endorsement of facilitators and barriers to the identification, management and referral of childhood anxiety disorders

| Item | % 'agree' + 'completely agree' | 95% CI | | % 'disagree' + 'completely disagree' | 95% CI | |
|---|---|---|---|---|---|---|
| **Identification of anxiety disorders in children under 12 years** | | Lower | Upper | | Lower | Upper |
| *Facilitators* | | | | | | |
| Feel confident in ability | 53 | 50.78 | 57.06 | 12 | 9.91 | 13.99 |
| Believe training was adequate | 21 | 19.09 | 24.29 | 48 | 44.68 | 50.98 |
| Comfortable broaching idea of anxiety disorders with the child | 80 | 77.72 | 82.74 | 7 | 5.14 | 8.30 |
| Comfortable broaching idea of anxiety disorders with the family | 88 | 86.81 | 90.79 | 3 | 1.92 | 4.08 |
| Believe it is part of a GP's responsibility | 93 | 91.60 | 94.78 | 1 | 0.54 | 1.93 |
| *Endorsement of barriers* | % 'quite a lot' + 'very much' | | | % 'a little' + 'not at all' | | |
| Limitations in children's communication abilities | 21 | 19.05 | 24.24 | 33 | 29.93 | 35.84 |
| Misinformation from parents | 27 | 24.06 | 29.65 | 35 | 32.52 | 38.55 |
| Concerns about stigmatising the child | 10 | 8.68 | 12.56 | 69 | 66.77 | 72.56 |
| Family concerns about stigma | 16 | 13.60 | 18.21 | 60 | 57.25 | 63.41 |
| Cultural barriers | 17 | 14.46 | 19.17 | 58 | 54.89 | 61.11 |
| Language barriers | 18 | 15.83 | 20.69 | 64 | 61.38 | 67.41 |
| Family reluctance to accept disorder | 20 | 17.73 | 22.80 | 43 | 39.80 | 46.03 |
| Time restrictions | 67 | 63.91 | 69.84 | 14 | 12.33 | 1.677 |
| Lack of available/accessible treatment | 68 | 98.34 | 99.61 | 15 | 12.91 | 17.42 |
| Lack of effective treatment | 45 | 42.53 | 48.80 | 27 | 23.91 | 29.48 |
| **Management of children under 12 years with anxiety disorders** | % 'agree' + 'completely agree' | | | % 'disagree' + 'completely disagree' | | |
| *Facilitators* | | | | | | |
| Feel confident in ability | 13 | 11.32 | 15.64 | 51 | 48.71 | 55.02 |
| Believe training was adequate | 10 | 8.29 | 12.09 | 66 | 63.56 | 69.50 |
| Comfortable discussing management strategies with the family | 52 | 49.33 | 55.63 | 20 | 17.32 | 22.34 |
| Comfortable discussing management strategies with the child | 46 | 43.29 | 49.58 | 22 | 19.21 | 24.42 |
| Aware of resources to aid families | 24 | 21.54 | 26.95 | 75 | 73.05 | 78.46 |
| Believe having a relationship with a family is beneficial | 94 | 92.54 | 95.52 | <1 | 0.39 | 1.66 |
| Aware of local agencies to support children and their families | 65 | 63.06 | 69.03 | 17 | 15.09 | 19.89 |
| Believe it is part of a GP's role | 69 | 66.70 | 72.49 | 10 | 8.01 | 11.77 |
| *Endorsement of barriers* | % 'quite a lot' + 'very much' | | | % 'a little' + 'not at all' | | |
| Cultural barriers | 14 | 12.01 | 16.43 | 60 | 58.72 | 64.86 |
| Language barriers | 16 | 13.98 | 18.65 | 64 | 61.32 | 67.37 |
| Time restrictions | 72 | 70.15 | 75.76 | 11 | 9.68 | 13.73 |
| Family reluctance to accept the disorder | 19 | 16.78 | 21.76 | 46 | 43.38 | 49.68 |
| Limitations in children's communication abilities | 18 | 15.80 | 20.67 | 43 | 40.60 | 46.86 |
| Misinformation from parents | 18 | 15.55 | 20.40 | 50 | 47/78 | 54.09 |
| Concerns about stigmatising the child | 7 | 5.91 | 9.25 | 71 | 69.31 | 74.98 |
| Family concerns about stigma | 11 | 9.23 | 13.21 | 63 | 60.58 | 66.66 |
| **Referral of children under 12 years with anxiety disorders** | % 'agree' + 'completely agree' | | | % 'disagree' + 'completely disagree' | | |

**Table 2** Continued

| Item<br><br>Identification of anxiety disorders in children under 12 years | % 'agree' + 'completely agree' | 95% CI | | % 'disagree' + 'completely disagree' | 95% CI | |
|---|---|---|---|---|---|---|
| | | Lower | Upper | | Lower | Upper |
| *Facilitators* | | | | | | |
| Long waiting times reduce the likelihood of making a referral | 50 | 47.37 | 53.66 | 35 | 32.65 | 38.68 |
| Parental pressure increases the likelihood of making a referral | 85 | 83.18 | 87.62 | 5 | 4.02 | 6.87 |
| Have a relationship with local specialist services | 6 | 4.30 | 7.24 | 79 | 76.49 | 81.62 |
| Believe that specialist services' interventions are effective | 78 | 75.63 | 80.83 | 4 | 3.22 | 5.84 |
| *Endorsement of barriers* | % 'quite a lot' + 'very much' | | | % 'a little' + 'not at all' | | |
| Long waiting times | 89 | 87.03 | 90.97 | 3 | 1.91 | 4.04 |
| Cultural barriers | 8 | 6.22 | 9.65 | 76 | 75.11 | 80.38 |
| Language barriers | 8 | 6.83 | 10.40 | 77 | 76.49 | 81.66 |
| Time restrictions | 30 | 28.26 | 34.14 | 51 | 49.40 | 55.73 |
| Family reluctance to accept the disorder | 12 | 10.47 | 14.68 | 62 | 60.68 | 66.78 |
| Limitations in children's communication abilities | 6 | 4.91 | 8.03 | 70 | 68.19 | 73.93 |
| Misinformation from parents | 9 | 7.83 | 11.59 | 66 | 64.11 | 70.06 |
| Concerns about stigmatising the child | 6 | 4.55 | 7.57 | 78 | 76.94 | 82.06 |
| Family concerns about stigma | 8 | 6.71 | 10.24 | 73 | 71.80 | 77.32 |
| Lack of available/accessible treatment | 54 | 51.78 | 58.10 | 28 | 25.49 | 31.22 |

**Table 3** Regression model of factors associated with a GP's confidence identifying childhood anxiety disorders

| | B | Exp(B) | 95% CI for Exp(B) | |
|---|---|---|---|---|
| | | | Lower | Upper |
| GP | | | | |
| Gender | 0.352* | 1.423 | 1.090 | 1.856 |
| Psychiatry training | 0.147 | 1.158 | 0.889 | 1.510 |
| Paediatric training | 0.155 | 1.168 | 0.861 | 1.5834 |
| Years qualified | −0.004 | 0.996 | 0.980 | 1.012 |
| Child over 5 years | 0.163 | 1.177 | 0.841 | 1.656 |
| Practice | | | | |
| No. of patients | 0.000 | 1.000 | 1.000 | 1.000 |
| Socio-economic status (deprivation decile of practice) | 0.062* | 1.064 | 1.006 | 1.125 |
| % non-white ethnic groups | 0.008 | 1.008 | 0.995 | 1.021 |
| Constant | 0.081 | 0.922 | | |

*$R^2$=0.01 (Hosmer & Lemeshow). 0.01 (Cox & Snell). 0.03 (Nagelkerke). Model $\chi^2(1)$=17.4, p<0.05.
GP, general practitioner.

**Table 4** Regression model of factors associated with a GP's confidence managing childhood anxiety disorders

| | B | Exp(B) | 95% CI for EXP(B) | |
|---|---|---|---|---|
| | | | Lower | Upper |
| GP | | | | |
| Gender | −0.239 | 0.787 | 0.531 | 1.168 |
| Psychiatry training | 0.273 | 1.314 | 0.887 | 1.947 |
| Paediatric training | 0.218 | 1.244 | 0.766 | 2.019 |
| Years qualified | 0.002 | 1.002 | 0.979 | 1.027 |
| Child over 5 years | 0.687* | 1.987 | 1.132 | 3.489 |
| Practice | | | | |
| No. of patients | 0.000 | 1.000 | 1.000 | 1.000 |
| Socio-economic status (deprivation decile of practice) | 0.090* | 1.094 | 1.004 | 1.193 |
| % non-white ethnic groups | 0.015 | 1.015 | 0.997 | 1.034 |
| Constant | −1.847 | .158 | | |

*$R^2$=0.05 (Hosmer & Lemeshow). 0.02 (Cox & Snell). 0.04 (Nagelkerke). Model $\chi^2(1)$=20.23, p=0.01.
GP, general practitioner.

chiatric and paediatric training as well as years of experience and gender did not have any impact on their confidence, nor did size or ethnicity of their practice population.

The table demonstrates that although GPs feel that it is their responsibility and feel comfortable discussing anxiety disorders with families, inadequate training as well as time restrictions and a lack of accessible treatment pose the biggest barriers for GPs in identification of childhood anxiety disorders. Lack of confidence, training, resources and time were the biggest barriers to management. GPs believed that specialist services were effective but waiting times posed the largest barrier to access and less than 10% of GPs felt that they had a relationship with these services. Stigma, language and culture issues were not seen as barriers to GPs' identification, management or referral.

## DISCUSSION
### Summary
Although most GPs believe that it is their responsibility to identify and manage anxiety disorders in children, only half of those surveyed felt confident identifying, and only 13% felt confident managing, these disorders. These figures may not be surprising given that only 21% of GPs believed their training in identification was adequate, only 10% believed that their training in management of anxiety disorders was adequate and the vast majority were not aware of any resources to provide families. Time restrictions also posed a barrier for both identification and management. Although 78% of GPs believed that specialist services' interventions were likely to be effective, long waiting times and lack of accessible treatments were overwhelmingly endorsed as a barrier to accessing specialist services and 79% felt that they did not have a relationship with such services. Free-text comments for this section highlighted that GPs felt CAMHS referral criteria were unclear, and often had very high thresholds. However, notably 85% indicated that referral was significantly increased by parental pressure, demonstrating the need for parents to be strong advocates for their children in order to access specialist services. Interestingly, although previous literature reported stigma surrounding mental health as a barrier for GPs, this was not the case for the GPs in this research.

Female GPs and those in less deprived practices were associated with more confidence identifying childhood anxiety disorders. Notably, GPs who had completed psychiatric or paediatric training were not associated with more confidence in identification, reinforcing the commonly reported view that the training received was inadequate. In terms of management, GPs in less deprived practices and those who were parents of a child over the age of 5 were associated with more confidence. This latter finding was particularly striking and was consistent with our previous qualitative work that suggested that GPs rely on their own personal experience, as their professional training was largely viewed as inadequate to manage these disorders.[28]

### Strengths and limitations
One of the key strengths of the study is the inclusion of a large, nationally representative group of GPs. Another is that it used findings from a recent systematic review,[23] qualitative study[28] and GP feedback to inform the development of the questionnaire, ensuring that the questions were meaningful to the GPs and that the most pertinent questions were asked. This appeared to have been successful as the free-text responses mainly reinforced the findings from the main questionnaire and little new information was added. Unlike many previous studies, the questionnaire distinguished between different stages of the primary care process (ie, identification, management, referral to specialist services). This allows pinpointing of where issues are most problematic, for example, time restrictions were a barrier for identification and management, but much less so for referral. We can also see that GPs feel significantly more confident identifying anxiety disorders than managing them. Future studies would benefit from examining these stages at a more granular level, for example, to further understand the skills and confidence necessary at different stages of the process, for example, identifying the problem, broaching the subject with the family/young person.

These strengths need to be considered in light of various limitations. Efforts were made to recruit a number of GPs to pilot the survey; however, only one responded, which is a limitation of the study. Although the sample had similar demographic characteristics to GPs across England and we recruited a substantial proportion of GPs (n=443; 46%) who did not consider themselves 'research active', no data were available on those who chose not to participate and we cannot exclude the possibility of a self-selection bias influencing the findings. We also did not assess GPs' abilities to accurately identify anxiety disorders in children and cannot rule out the possibility that in some cases GPs answered questions in relation to mental health disorders other than anxiety. This survey involved GPs reporting on their own experiences, which may be influenced by poor recall and social desirability biases. Furthermore, as this survey is cross-sectional, conclusions cannot be drawn about causality. Most free-text comments related to questions asked in the survey; however, an additional area that GPs raised in the comments was the role of others in identification of anxiety disorders and for referrals to specialist services, in particular, schools and health visitors. A final limitation of this study is the lack of distinction between anxiety disorders, which are not a homogenous group, as well as the fact that we focused on anxiety disorders as a distinct condition, despite the fact that they are often comorbid with other psychological and physical health disorders. This may limit the generalisability of this study.

### Comparison with previous research
The specific focus on anxiety disorders is a further strength of this research given the specific challenges

faced with this condition, and this contrasts with much of the previous work in this area which does not distinguish between mental health disorders. Many of the findings in this study echo previous qualitative findings,[24 28] such as the positive impact of being a parent on confidence managing child anxiety disorders and issues associated with long waiting times and poor access to specialist services. It also reinforced the idea that GPs working in more deprived areas may experience particular difficulties potentially due to more complex family situations. Future studies would benefit from a more in-depth examination into the association between deprivation and GP confidence in identification and management. However, although some of the GPs interviewed in the previous research seemed to have some uncertainty regarding their role in the identification and management of this condition, it appeared that the majority of GPs surveyed believed that this is their responsibility, at least in part. This research also reinforced a recent stem4 report which found that GPs had serious concerns about accessing specialist help for young people with mental health disorders.[32] The issue of poor access to services and long waiting times are common across the mental health literature, but this study highlights a number of novel findings, such as the widespread lack of confidence in management of childhood anxiety disorders, and the stronger role of personal experience of being a parent than access to relevant specialist training in GP confidence managing childhood anxiety disorders.

### Implications for clinicians, policy-makers and researchers

The focus of this research was to investigate barriers, from a GP's perspective; however, in order to fully understand how to increase access to appropriate support for childhood anxiety disorders, future studies could seek to capture objective data (eg, referral data), in order to corroborate the questionnaire findings. This would be useful to understand whether self-report correlates with actual behaviour. Furthermore, providing GPs with vignettes describing children with and without anxiety disorders may be useful to determine whether a GP's perception of their confidence is related to their actual identification abilities.

This research strongly suggests that GPs' current training in childhood anxiety disorders is inadequate, and highlights the need to place increased emphasis on training, both in medical school training and beyond, to boost confidence and develop skills. This reinforces previous qualitative findings which highlight that GPs believed that their training in mental health problems had been inadequate, and focused on adults to the detriment of children and young people.[28] This training should be meaningful for GPs and firmly grounded in the context of primary care, as well as focused specifically on child and adolescent populations. This supports recommendations from the Royal College of General Practitioners that GPs should receive specialist-led training in child health and mental health problems.[33] It is encouraging that the

majority of GPs surveyed see identifying and managing this condition as part of their role, in addition to other organisations (eg, schools), but it is possible that a lack of confidence and comfort with the topic matter prevents GPs from fully engaging with a condition.[34] In addition to further training, previous qualitative research suggests that increased contact and collaboration with specialist services would increase GPs confidence in this area.[28] Furthermore, as with previous studies in this area, difficulty accessing specialist services in a timely manner poses a significant barrier for GPs who commonly used the free-text comments to express the view that CAMHS' referral thresholds are prohibitively high, and service inclusion criteria are unclear. Increasing GPs confidence in this area, as well as providing them with more effective resources, may encourage and enable GPs to manage many cases in primary care and only refer severe or complex cases, thus alleviating pressure on specialist services.

## CONCLUSION

Failing to intervene effectively with childhood anxiety disorders brings negative implications for both individual children and families[6] and increased costs for society.[8] GPs consider identification and management of childhood anxiety disorders to be within their role; however, they face a number of barriers and lack confidence in their abilities. Increased GP training in identification and management of child anxiety disorders may help to strengthen GPs' confidence and ability to effectively support children under the age of 12 with anxiety disorders. Furthermore, the findings suggest that GPs would benefit from increased access to ongoing training and appropriate resources to support identification and early management and to facilitate better communication around access to specialist services for children with anxiety disorders.

**Acknowledgements** We would like to acknowledge the support of the following National Institute for Health Research Clinical Research Networks: Thames Valley & South Midlands, East Midlands, Eastern, North West London, South London, North East and North Cumbria, South West Peninsula, West Midlands, Yorkshire & Humber, North Thames, Wessex. We would also like to thank all of the General Practitioners who participated in this research.

**Contributors** All three authors were involved in the design of this study. Participant recruitment and analysis was conducted by DOB. The main draft of the manuscript was written by DOB with input by KH and CC.

**Funding** This work was supported by grant from the National Institute for Health Research 'Research Professorship' to CC (NIHR-RP-2014-04-018).

**Disclaimer** The views expressed are those of the authors and not necessarily those of the NHS, the NIHR or the Department of Health.

**Competing interests** None declared.

**Patient consent for publication** Not required.

**Ethics approval** School of Psychology & Language Science Ethics Board at the University of Reading. Approval was also granted by the Health Research Authority to conduct the study with NHS staff.

**Provenance and peer review** Not commissioned; externally peer reviewed.

**Data sharing statement** Full data set and statistical codes are available from the corresponding author. Data supporting the results reported in this paper are openly available from the University of Reading Research Data Archive at http://dx.doi.org/10.17864/1947.192.

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
