## [Reviewer comments · BMJ Open]

ARTICLE DETAILS

TITLE (PROVISIONAL)	Barriers and facilitators to identification and management of childhood anxiety disorders in primary care: a survey of General Practitioners in England
AUTHORS	O'Brien, Doireann; Harvey, Kate; Creswell, Cathy

VERSION 1 - REVIEW

REVIEWER	Sonja March University of Southern Queensland, Australia Have co-authored a paper with one of the authors in the last 12 months.
REVIEW RETURNED	08-Aug-2018

GENERAL COMMENTS	Thank you for the opportunity to review this article "Barriers and facilitators to identification and management of childhood anxiety disorders in primary care: a survey of general practitioners". This article was interesting, well written and focused on an important topic. Understanding the barriers and facilitators to the identification and management of childhood anxiety is crucial, and especially so in GPs who are the first line of service delivery, not only in the UK but world wide. This article has relevance and implications outside the UK and provides important information regarding the barriers that might prevent children from being identified and receiving appropriate help. The authors presented the results clearly and provided excellent discussion of the strengths and limitations of this research. The outcomes of this research will be useful in informing clinical practice and translational research regarding health service delivery to children with anxiety (and beyond). I have only minor comments and questions regarding this article. In particular, a strength of this study was the examination of three separate processes, identification, management and referral, as blockages can occur at any of these processes and clearly are different. I was interested to know whether the authors considered examining Identification at a more granular level? For example, I wonder whether GPs would demonstrate different confidence and ability in various aspects of identification, eg. identification of a general anxiety problem through standard consultation vs administration of formal and informal assessments to assess if anxiety exists, interpretation of assessments, gathering of clinical information through interview with child/parent/teacher etc. I imagine there would be a distinct difference between those GPs who sense there is anxiety and don't know what to do with it, versus those who can conduct basic assessments, and therefore
---

	are armed with more information to facilitate referral. This would also assist in informing future training interventions. I also wonder whether the authors gathered any data on whether the reported barriers actually prevented GPs from referring, or whether they simply made it more difficult? Did the authors gather any qualitative or other data here, or in their other studies, regarding GP's thoughts on how this situation can be improved? It would be useful to know how GP's felt they could enhance their skills and confidence in this area. The authors comment in the discussion that additional training may not be the answer, and it would be useful to gather GP data on this - if this has been collected in other studies, it would be useful to refer to it in the discussion. Finally, I wonder whether the authors could comment more in the discussion regarding the predictors of confidence in identification and management. Specifically, the finding that being in less deprived areas meant higher confidence was interesting. Do the authors have thoughts on why this is the case? Could it be that child presentations in higher deprivation areas might be more complex, severe or part of more dysfunctional family contexts? Does this indicate that identification and management in these circumstances go beyond a GPs ability? Or that further training is required here especially in referral to appropriate services? I enjoyed reading this article and believe it provides valuable information for clinicians and researchers.
--	---

REVIEWER	Xinyu Zhou Department of Psychiatry, The First Affiliated Hospital of Chongqing Medical University, Chongqing, China
REVIEW RETURNED	10-Aug-2018

GENERAL COMMENTS	The authors are to be congratulated for this interesting study. The following suggestions may make the results more understandable for readers: The introduction about the general practitioners in UK is too lengthy. Please streamlined these word. The authors could expand on this statement in their discussion: "Unlike many previous studies, the questionnaire distinguished between different stages of the primary care process (i.e. identification, management, referral to specialist services)." As the author said that "A final limitation of this study is the lack of distinction between anxiety disorders, which are not a homogenous group, as well as the fact that we focused on anxiety disorders as a distinct condition, despite the fact that they are often co-morbid with other psychological and physical health disorders", most young patients with anxiety disorders often co-morbid with depressive disorders; thus the general practitioners may misdiagnosis of anxiety disorders. Please clarify this issue.
---

	The results contain many correlations and it is a bit confusing to the reader. I would suggest highlighting only the most significant or important and summarizing the others in tables. The most important findings should be the ones discussed. There are some small errors, such as the “managing” in the line 194 may be deleted, please check again.
--	--

REVIEWER	Patricia Moreno-Peral Institute of Biomedical Research in Malaga (IBIMA). Spain.
REVIEW RETURNED	14-Aug-2018

GENERAL COMMENTS	This study aimed to determine the barriers to and facilitators for identifying, managing and accessing specialist services for anxiety disorders in pre-adolescent children from General Practitioners in England; as well as factors associated with GPs' confidence. The article is well written and addressed an interesting topic, well justified. Additionally, the authors evaluate a large sample of GPs in England. However, I have some questions and comments that the authors may want to clarify:  - The factors that predict GPs' confidence in identifying and managing childhood anxiety disorders cannot be determined through a cross-sectional study. It would be more appropriate to refer to an association. This is especially important in the factors or variables referring to the GP's practice. - The authors point out that corresponding, qualitative interviews in this study were carried out. Were the interviews performed with the same GPs with the aim of carrying out some kind of triangulation, or were they simply two different complementary studies? - What about the reliability of the questionnaire? A reliability analysis is missing. At a minimum, the test-retest reliability of the questions could have been evaluated. - Although there was little missing data, multiple imputations would have been the most appropriate way to address missing data. I think this should be addressed in the limitations of the study. - How were the two dependent variables on the two regression models constructed (Tables 3 and table 4)? Were they constructed from the variable “feel confidence in ability”? This aspect needs clarification. - Regarding factors or independent variables, why were those 8 factors selected and included in the regression model? - A new paragraph in the results comparing barriers and facilitators in identification, management and referral would be highly welcome. - With respect to the limitations, the authors should include the limitations regarding the design of the study, the non-performance of multiple imputations, and the limitations of the questionnaire itself.
--

VERSION 1 – AUTHOR RESPONSE

Reviewer: 1

In particular, a strength of this study was the examination of three separate processes, identification, management and referral, as blockages can occur at any of these processes and clearly are different. I was interested to know whether the authors considered examining Identification at a more granular level?

We agree with the reviewer that it would indeed have been very interesting and informative to consider each stage of the process at a more granular level. As it was, in this study, there were already a large number of questions included in the survey and we were reluctant to include more. We have added a line in the manuscript to highlight that future studies would benefit from further study in this area. We thank you for your query and hope that this response is satisfactory.

Future studies would benefit from examining these stages at a more granular level, for example, to further understand the skills and confidence necessary at different stages of the process e.g. identifying the problem, broaching the subject with the family/young person. I also wonder whether the authors gathered any data on whether the reported barriers actually prevented GPs from referring, or whether they simply made it more difficult?

In this study we only report GP self-reported barriers, i.e. their perceived difficulty. As we note in the discussion (p 17) future studies would benefit from accessing administrative data showing number and frequency of referrals by GPs.

Did the authors gather any qualitative or other data here, or in their other studies, regarding GP's thoughts on how this situation can be improved? It would be useful to know how GP's felt they could enhance their skills and confidence in this area. The authors comment in the discussion that additional training may not be the answer, and it would be useful to gather GP data on this - if this has been collected in other studies, it would be useful to refer to it in the discussion.

We did not state that 'additional training may not be the answer', rather one of the key recommendations of this study is to strengthen and improve the training that GPs currently receive. However, we agree with the reviewer that GP's views on the utility of further (or different) training would be of benefit. We did explore this in a previous qualitative study and GPs expressed differing opinions on this. Almost universally, GPs believed that their training in mental health problems had been inadequate, and focused on adults to the detriment of children and young people. We have added a line expressing this to the paper.

This reinforces previous qualitative findings which highlight that GPs believed that their training in mental health problems had been inadequate, and focused on adults to the detriment of children and young people(33).

We have also added a line to the discussion to provide data regarding other suggestions GPs had for how the situation could be improved.

In addition to further training, previous qualitative research suggests that increased contact and collaboration with specialist services would increase GPs confidence in this area (33). Finally, I wonder whether the authors could comment more in the discussion regarding the predictors of confidence in identification and management. Specifically, the finding that being in less deprived areas meant higher confidence was interesting. Do the authors have thoughts on why this is the case? Could it be that child presentations in higher deprivation areas might be more complex, severe or part of more dysfunctional family contexts? Does this indicate that identification and management in these circumstances go beyond a GPs ability? Or that further training is required here especially in referral to appropriate services?

We agree that this is an interesting finding. We explored this topic in our previous qualitative study and have added some lines to the manuscript to discuss this further, and to make stronger recommendations in this regard, as well as recommending that future studies explore this association in greater depth.

It also reinforced the idea that GPs working in more deprived areas may experience particular difficulties potentially due to more complex family situations. Future studies would benefit from a more in-depth examination into the association between deprivation and GP confidence in identification and management

I enjoyed reading this article and believe it provides valuable information for clinicians and researchers.

Many thanks for your comments and for taking the time to review this article.

Reviewer: 2

The introduction about the general practitioners in UK is too lengthy. Please streamline these words.

We thank you for your comment. We have streamlined some of this text regarding GPs in the UK in the introduction. N.B. We have retained reference to UK data as this sets the context for this particular (UK-based) study.

As such, they are well placed to support families, and are as often seen as “gate-keepers” to accessing mental health services

...feel that they need further training, particularly in the development of mental health disorders in childhood and adolescence as their current provision is very focused on adults

In addition, they face barriers such as insufficient resources when managing mental health conditions in primary care and extensive waiting lists for specialist services(23) and a high rate of rejected referrals. Although GPs are the professional group who make the most referrals to Child and Adolescent Mental Health Services (CAMHS), the likelihood that their referrals will be rejected is very high.

The reason for this is unclear, but possibly from the corresponding qualitative interviews in this study reflects issues with GP detection of mental health problems in this age group and lack of clarity regarding thresholds for CAMHS amongst GPs, associated with a lack of decision making aids for child and adolescent mental health disorders.

The authors could expand on this statement in their discussion:

"Unlike many previous studies, the questionnaire distinguished between different stages of the primary care process (i.e. identification, management, referral to specialist services)."

Thank you for this comment. However, we believe that the subsequent sentence ‘This allows pinpointing of where issues are most problematic, for example, time restrictions were a barrier for identification and management, but much less so for referral. We can also see that GPs feel significantly more confident identifying anxiety disorders than managing them’ expands on this statement.

As the author said that “A final limitation of this study is the lack of distinction between anxiety disorders, which are not a homogenous group, as well as the fact that we focused on anxiety disorders as a distinct condition, despite the fact that they are often co-morbid with other psychological and physical health disorders”, most young patients with anxiety disorders often co-morbid with depressive disorders; thus the general practitioners may misdiagnose anxiety disorders. Please clarify this issue.

We have added the following line to the study, which will hopefully clarify the issue:

A final limitation of this study is the lack of distinction between anxiety disorders, which are not a homogenous group, as well as the fact that we focused on anxiety disorders as a distinct condition, despite the fact that they are often co-morbid with other psychological and physical health disorders. This may limit the generalisability of this study.

The results contain many correlations and it is a bit confusing to the reader. I would suggest highlighting only the most significant or important and summarizing the others in tables. The most important findings should be the ones discussed.

We have attempted to do as you have suggested and have summarised the results pulling out the most important findings in the text in the 'results' section (lines 244-251). We have also attempted to summarise and draw attention to the most important results in the discussion (lines 284-299).

There are some small errors, such as the "managing" in the line 194 may be deleted, please check again.

Thank you for this comment, we have rechecked the document and made changes.

Reviewer: 3

- The factors that predict GPs' confidence in identifying and managing childhood anxiety disorders cannot be determined through a cross-sectional study. It would be more appropriate to refer to an association. This is especially important in the factors or variables referring to the GP's practice.

Thank you for this comment. We have changed the terminology in the results section of the manuscript, referring to 'an association' rather than a 'prediction' and have added a line to the limitations to prevent any implication of causality.

Furthermore, as this survey is cross-sectional, causality cannot be implied.

- The authors point out that corresponding, qualitative interviews in this study were carried out. Were the interviews performed with the same GPs with the aim of carrying out some kind of triangulation, or were they simply two different complementary studies?

Apologies for any confusion, these were two different complementary studies. We have checked the manuscript and feel that this is clear but please let us know if you disagree.

- What about the reliability of the questionnaire? A reliability analysis is missing. At a minimum, the test-retest reliability of the questions could have been evaluated.

As this is a survey and is not intended to be used as a standardised instrument, the psychometric properties of the questionnaire were not examined (in line with other papers published in this journal, e.g.¹²³). For example, we would not expect the survey, which was intended for a single-use purpose, to be measuring a stable characteristic that would have high test-retest reliability.

- Although there was little missing data, multiple imputations would have been the most appropriate way to address missing data. I think this should be addressed in the limitations of the study.

Given how little missing data there was in this survey (i.e. between 7 to 20 cases in the 971 respondents, representing a very low percentage – approx. 2%) we do not believe that it is necessary to conduct multiple imputations. We hope that you find this response satisfactory.

- How were the two dependent variables on the two regression models constructed (Tables 3 and table 4)? Were they constructed from the variable “feel confidence in ability”? This aspect needs clarification.

We have added some more explanation around this, which hopefully clarifies the methods.

To address research question 2, logistic regression analyses were conducted with the dependent variables of confidence a) identifying² and b) managing² anxiety disorders in children under 12 years. These dependent variables were derived from the questions: ‘I am confident identifying anxiety disorders’ and ‘I am confident managing anxiety disorders’. These questions were transformed into a binary format by collapsing; ‘agree’ and ‘completely agree’ together and ‘neutral’, ‘disagree’ and ‘completely disagree’ together.

- Regarding factors or independent variables, why were those 8 factors selected and included in the regression model?

We have added in a line to address the reason for choosing these factors.

These factors were chosen based on previous qualitative research which indicated the potential influence they may have on a GP’s confidence (33).

- A new paragraph in the results comparing barriers and facilitators in identification, management and referral would be highly welcome.

Thank you for this comment. We believe that the paragraph from lines 244-251 compares the barriers in identification, management and referral. Currently, wordcount limitations prevent us from elaborating on this further, however, if the wordcount is more flexible, we are happy to present a longer discussion.

- With respect to the limitations, the authors should include the limitations regarding the design of the study, the non-performance of multiple imputations, and the limitations of the questionnaire itself.

We believe that we have addressed the main limitations of this study, including questionnaire (only one GP to pilot, lack of objective data to triangulate, possibility of self-selection bias). However, we have added one further limitation and if there are further limitations that you believe we should include, please do make us aware.

This survey involved GPs reporting on their own experiences, which may be influenced by poor recall and social desirability biases. Furthermore, as this survey is cross-sectional, conclusions cannot be drawn about causality.

VERSION 2 – REVIEW

REVIEWER	Patricia Moreno Peral Biomedical Research Institute of Malaga (IBIMA)
REVIEW RETURNED	13-Nov-2018

GENERAL COMMENTS	The authors have addressed my comments. Congratulations for this work.
--